# Comparison of 6-week PMTCT outcomes for HIV-exposed and HIV-unexposed infants in the era of lifelong ART: Results from an observational prospective cohort study

Appolinaire Tiam[1,2]*, Seble G. Kassaye[3], Rhoderick Machekano[1], Vincent Tukei[4], Michelle M. Gill[1], Majoalane Mokone[4], Mosilinyane Letsie[5], Mots'oane Tsietso[5], Irene Seipati[5], Janety Barasa[4], Anthony Isavwa[4], Thorkild Tylleskär[2], Laura Guay[1,6]

1 Elizabeth Glaser Pediatric AIDS Foundation, Washington D.C., United States of America, 2 Centre for International Health, University of Bergen, Bergen, Norway, 3 Department of Medicine Georgetown University School of Medicine, Washington D.C., United States of America, 4 Elizabeth Glaser Pediatric AIDS Foundation, Maseru, Lesotho, 5 Ministry of Health, Maseru, Lesotho, 6 Department of Epidemiology and Biostatistics, George Washington University Milken Institute School of Public Health, Washington D.C., United States of America

* atiam@pedaids.org

**Data Availability Statement:** All relevant data are within the manuscript and its Supporting Information files.

## Abstract

### Background

Lifelong antiretroviral therapy (ART) reduces mother-to-child HIV transmission (MTCT) and improves maternal health. Data on the outcomes of HIV-exposed infants (HEI) compared to their unexposed counterparts in the era of universal ART is limited. We compared birth and 6-week outcomes among infants born to HIV-positive and HIV-negative women in Lesotho.

### Methods

941 HIV-negative and 653 HIV-positive pregnant women were enrolled in an observational cohort to evaluate the effectiveness of prevention of mother-to-child HIV transmission (PMTCT) program after implementation of universal maternal ART in 14 health facilities. Pregnancy, delivery, birth, and 6-week data were collected through participant interviews and medical record review. DNA PCR testing for HEI was conducted within 2 weeks of birth and at around 6 weeks of age. Data were analysed to estimate the distribution of birth outcomes, mortality, HIV transmission and HIV-free survival at 6 weeks.

### Results

HIV-positive women were older (mean age of 28.7 vs. 24.4 years) and presented for antenatal care earlier (mean gestational age of 23.0 weeks vs 25.3 weeks) than HIV-negative women. Prematurity was more frequent among HEI, 7.8% vs. 3.6%. There was no difference in rates of congenital anomalies between HEI (1.0%) and HIV-unexposed infants (HUI) (0.6%). Cumulative HIV transmission was 0.9% (N = 4/431) (95% CI:0.25–2.36) at birth and 1.0% (N = 6/583) (95% CI:0.38–2.23) at 6 weeks. Overall mortality, including

**Funding:** This work was made possible by the United States Agency for International Development (USAID) and the generous support of the American people through USAID Cooperative Agreement Number 674-A-00-10-00031-00 and No. AID-674-A-16-00005. The content included here is the responsibility of the authors and does not necessarily represent the official views of these donors.

**Competing interests:** The authors have declared that no competing interests exist

stillbirths, was 5.2% and 6.0% by 6 weeks for HUI and HEI respectively. Among liveborn infants, 6-week HIV-free survival for HEI was 95.6% (95% CI:93.7–97.1) compared to 96.8% (95% CI:95.4–97.9) survival for HUI.

## Conclusions

Implementation of universal maternal ART lowers MTCT at 6 weeks of age with no differences in congenital anomalies or early mortality between HIV exposed Infants and HIV unexposed infants. However, HIV exposed infants continue to have high rates of prematurity despite improved maternal health on ART.

## Introduction

Effective prevention of mother to child transmission (PMTCT) programs offering universal life-long antiretroviral therapy (ART) reduce HIV transmission to children from their mother and improve maternal health [1,2]. A critical question remains as to whether the reduction in MTCT rates and improvement in maternal health are coupled with improvement in birth outcomes and survival among HIV-exposed infants (HEI) to match those of HIV-unexposed infants (HUI).

There have been conflicting data reported on birth outcomes among HIV-positive women who are on lifelong ART compared to HIV negative women. Some studies found that adverse birth outcomes, such as increased preterm deliveries, stillbirths and low birth weight, occurred more frequently among HEI [3–8]. Other studies found no association between use of ART and adverse birth outcomes [9,10].

Intrauterine and perinatal HIV transmission measurement in the era of lifelong ART is limited. In Rwanda, a study measuring HIV-free survival in a cohort of HEI, found a 6-week Mother to child transmission rate of 0.5% (95% CI:0.2–1.6) demonstrating the effectiveness of lifelong ART for HIV-positive pregnant women in preventing perinatal HIV transmission [11]. UNAIDS spectrum data in Lesotho reported an estimated 6-week transmission of 6.9% in 2016 [1].

Lesotho is a mountainous country in southern Africa with one of the highest HIV prevalence documented in the world. The Lesotho Ministry of Health implemented universal ART for all HIV-positive pregnant women in 2013 using a once-daily fixed dose combination of tenofovir disoproxil fumarate (TDF), lamivudine (3TC) and efavirenz (EFZ) [12,13]. HIV-exposed infants are given nevirapine (NVP) for 6 weeks after birth.

The roll out of lifelong ART for all pregnant women in Lesotho provided a unique opportunity to determine its effect on birth outcomes and survival of infants born to HIV-positive mothers compared with a similar cohort of HIV-negative mothers and their HIV-unexposed infants. The study also assessed the effectiveness of the Lesotho PMTCT program in reaching the goal of the virtual elimination of new pediatric HIV infections in selected sites in Lesotho.

## Methods

### Design

HIV-positive and HIV-negative pregnant women attending routine antenatal care (ANC) services were enrolled in a prospective observational cohort study from June 2014 to February 2016. Study personnel captured demographic, social, and medical information through

participant interviews and extraction of medical record information during pregnancy, and at delivery, birth and 6 weeks postpartum.

## Setting

The study was conducted in 3 districts in Lesotho–Botha-Bothe, Thaba-Tseka and Mohale's Hoek. These districts were selected to include areas with varying levels of PMTCT service delivery coverage, and heterogeneity in health-seeking behavior due to variances in terrain (lowlands, foothills, mountains). We included all 5 hospitals in these districts and randomly selected 9 medium volume (100–200 ANC women/year) or high (>200 ANC women/year) health centers to be included in the study.

## Population and enrollment

Eligible HIV-positive and HIV-negative pregnant women attending ANC in the study health facilities were recruited for this study. They were eligible for study enrollment if they resided in the area, planned to continue to receive services at the facility following delivery, were willing to have their infant co-enrolled after birth, and were willing to provide written informed consent. Population proportional sampling was used to estimate the enrollment target at each of the 14 study facilities. The sample size for HIV-positive women was calculated based on a 4% estimated HIV transmission at 6–8 weeks in Lesotho at the time of study initiation with a precision of 1.4% [12]. Consecutive consenting women were enrolled until the sample size was reached. HIV-negative women were enrolled into an HIV seroincidence cohort study with scheduled repeat HIV testing at 36 weeks gestation, delivery, and every 3–6 months postpartum until 24 months postpartum [14].

## Data collection

Study data were collected by trained study nurses through structured interviews and by abstraction of data from clinical and laboratory records. Mother-infant pair information such as demographic variables, date of last menstrual cycle, HIV status of spouse, disclosure of HIV status was collected through structured interviews with the pregnant women. General health and clinical history, ARV use and toxicity, adherence, retention in care, family planning, infant feeding practices and infant growth were extracted from the women medical charts and clinic registers. Electronic tablets were used to enter data directly into a web-based database (SurveyCTO).

Gestational age at birth was estimated by the time between the date of last menstrual period given by the women at first ANC and the date of birth. Very preterm birth was defined as infant born at a gestational age of 28–32 weeks while preterm birth was infant born after 32 weeks but before 37 weeks [15]. In addition, miscarriage was defined as loss of pregnancy before the gestational age of 28 weeks and stillbirth was considered when the pregnancy was lost after 28 weeks [16].

## Laboratory methods

Blood samples were collected from a subset of HEI at or within 2 weeks of birth and from all HEI at 6–8 weeks of age to determine their HIV infection status. Trained nurses obtained blood from infants by heel prick, which was spotted directly onto filter paper to create a dried blood spot for HIV DNA PCR testing as per the standard Ministry of Health (MOH) protocol. HIV testing was performed by the national reference laboratory using the Roche COBAS

Ampliprep/COBAS TaqMan HIV-1 qualitative test (v2.0). Test results were obtained directly from the laboratory and entered into the study database.

## Statistical analysis

Quantitative data analysis was performed using STATA version 15.1 (College Station, TX, USA). We summarized categorical variables using frequencies and percentages of participants and continuous variables using means (+/- standard deviation). Maternal baseline characteristics were stratified by HIV status at enrollment. We compared birth outcomes between HIV-exposed and HIV-unexposed infants. HIV-free survival was estimated as the proportion of children alive and HIV-negative among all exposed children. The precision around survival estimates was assessed by 95% confidence intervals. We used the Kaplan Meier curves to graphically display infant mortality, infection, and HIV free survival. We performed complete case analysis, and missing data were not imputed.

## Ethical considerations

The study was approved by the Lesotho Ministry of Health Research and Ethics Committee, the Baylor College of Medicine Children's Foundation Lesotho Institutional Review Board (IRB), and the George Washington University Committee on Human Research IRB. All women were informed of the study protocol, requirements, and risks and benefits, and provided written informed consent to participate in the study.

## Results

### Participant characteristics

A total of 1594 pregnant women (941 HIV-negative and 653 HIV-positive) were enrolled in the study with their infants (Fig 1). Eight HIV negative women seroconverted before delivery. Delivery information was available for 95.4% of HIV positive women (623/653 and 92.2% of HIV negative women (868/941). 623 HIV positive women gave birth to 631 HIV exposed infants (HEI) and 868 HIV negative women gave birth to 879 HIV unexposed infants (HUI). Six week-follow-up information was available for 577 and 831 HEI and HUI respectively.

Characteristics of HIV-positive and HIV-negative study women at enrollment are presented in Table 1. HIV-positive pregnant women were older than their HIV-negative counterparts with a mean age of 28.7 (+/- 5.5) compared to 24.4 (+/- 5.7) years. HIV-positive women also presented earlier for ANC at a mean gestational age of 23.0 (+/- 8.7) weeks compared to 25.3 (+/- 8.2) weeks. Overall, 83.5% of the women were married, and 59.3% had disclosed their HIV-status to their partner/husband. Concerning HIV status of spouses, 4.2% of HIV negative women had HIV positive partners while 29.6% of HIV positive women had an HIV negative partner.

Among HIV-positive women, 97.8% were receiving ART at enrollment with 84.2% receiving the TDF/3TC/EFV first-line regimen. Among the 619 women with data available on the timing of ART, 249 (40.2%) initiated ART before conception and 370 (59.8%) initiated ART after conception.

### Birth outcomes

Overall, 91.6% of study women delivered in a health facility and 96.8% (n = 1443) of infants were born alive (Table 2). There was no difference in the proportion of infant stillbirths, however, HIV-positive women were more likely to have had a macerated stillbirth (consistent with antepartum death), while HIV-negative women were more likely to have had an intrapartum

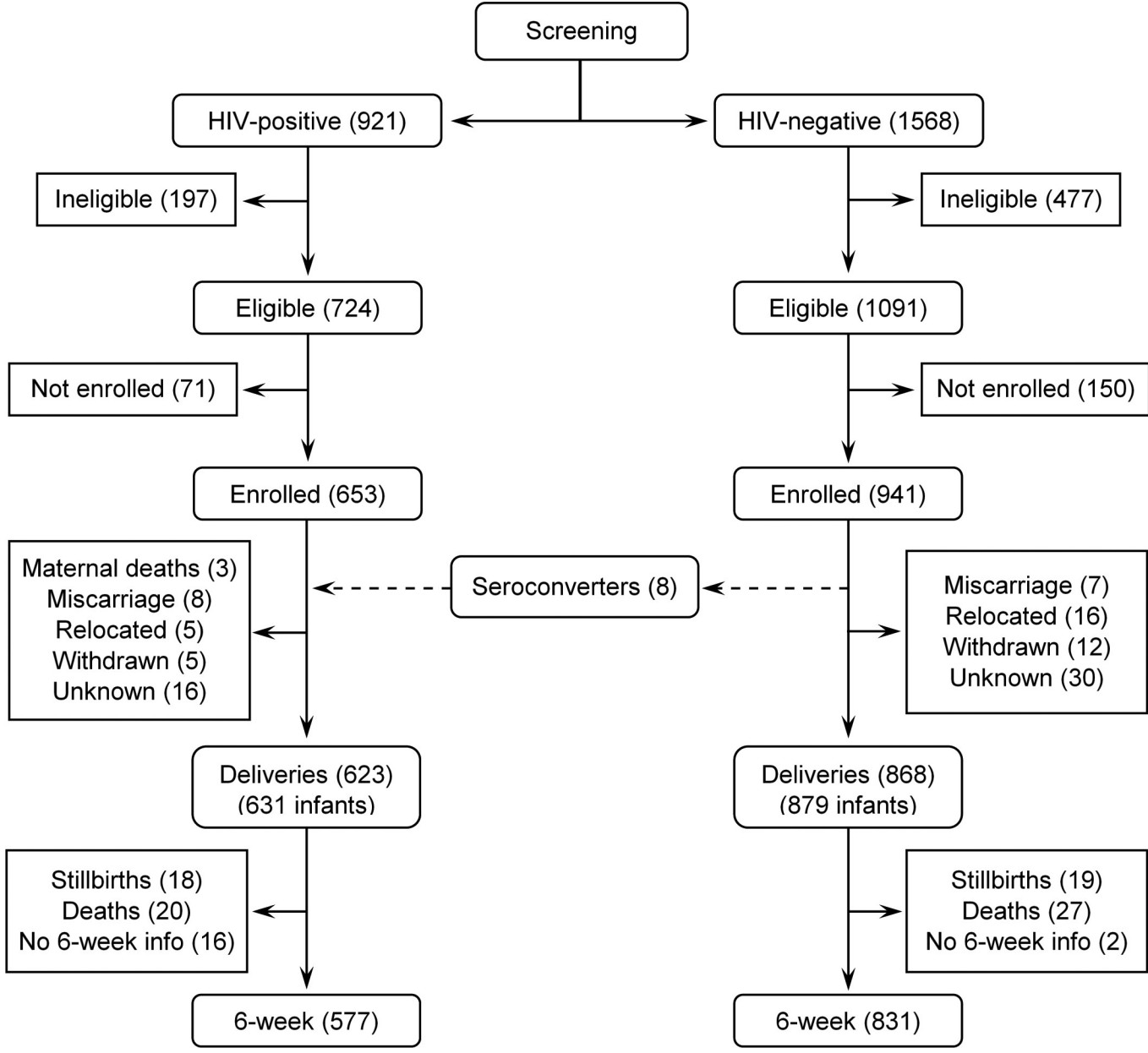

**Fig 1. Study enrollment for comparison of 6-week PMTCT outcomes for HEI and HUI in the era of lifelong ART.**

death (fresh stillbirth or a liveborn infant who died within two hours of delivery). The risk of premature birth in HEI was more than double the risk of prematurity in HUI (7.8% vs 3.6%). The risk of very premature births (<32 weeks gestational age) was also substantially higher in HEI (2.2%) compared to HUI (0.4%).

Of the 249 women who initiated ART pre-conception, 24 (9.6%) had premature babies compared to 27 (7.3%) among the 370 women who initiated ART post-conception. The rates of very low weight (<1.5kgs) and low birth weight (<2.5 kgs) among women who initiated ART before conception were 1.7% and 11.6% respectively compared to 1.2% and 11.9% among women who initiated ART after conception.

**Table 1. Characteristics of study women at enrollment.**

| | Maternal HIV Status | | Total Mothers |
|---|---|---|---|
| | HIV-negative (N = 941) | HIV-positive (N = 653*) | Total (N = 1593) |
| | n (%) | n (%) | n (%) |
| Maternal age in years (mean +/- SD) | 24.4 +/- 5.7 | 28.7 +/- 5.5 | 26.0 +/- 6 |
| Gestational age at first ANC in weeks (mean, SD) | 25.3 +/- 8.2 | 23.0 +/- 8.7 | 24.4 +/- 8.5 |
| Marital Status: Married | 801 (85.1) | 528 (81.0) | 1329 (83.4) |
| Mother disclosed HIV status to husband/partner | 521 (55.4) | 422 (64.9) | 943 (59.3) |
| Missing Data | 0 | 3 | 3 |
| **Maternal ARV Regimen at enrollment** | | | |
| TDF+3TC+EFV | N/A | 548 (84.2) | 548 (84.2) |
| TDF+3TC+NVP | N/A | 25 (3.8) | 25 (3.8) |
| AZT+3TC+EFV | N/A | 28 (4.3) | 28 (4.3) |
| AZT+3TC+NVP | N/A | 29 (4.5) | 29 (4.5) |
| ART-other regimens | N/A | 7 (1.1) | 7 (1.1) |
| None | N/A | 14 (2.2) | 14 (2.2) |
| Missing data | N/A | 1 | 1 |
| **Husband/Partner's HIV Status** | | | |
| Positive | 19 (4.2) | 230 (68.9) | 249 (31.8) |
| Negative | 422 (94.0) | 99 (29.6) | 521 (66.5) |
| Unknown | 8 (1.8) | 5 (1.5) | 13 (1.7) |
| Not tested | 492 | 318 | 810 |
| **Husband/Partner Taking ARVs** | | | |
| Yes | 14 (73.7) | 166 (72.2) | 180 (72.3) |
| No | 5 (26.3) | 63 (27.4) | 68 (27.3) |
| Unknown | 0 | 1 (0.6) | 1 (0.4) |

*One woman was enrolled but excluded from analysis due to missing enrolment questionnaire data

The rate of congenital anomalies was 0.6% and 1% among HEI and HUI respectively.

## Infant survival

There were no substantial differences in the rates of infant survival at 6 weeks of age by infant HIV exposure status (Fig 2 and Table 3). When including all deaths (liveborn plus stillbirths), the estimated survival rates were 94.8% (95% CI: 93.1–96.1) among HUI and 94.0% (95% CI: 91.8–95.7) among HEI. Analysis of postnatal deaths only (excluding stillbirths), yielded estimated survival rates of 96.8% (95% CI: 95.4–97.9) and 96.7% (95% CI: 95.0–98.0) for HUI and HEI respectively. Adjusting for maternal mortality and gestational age at first ANC visit, infant HIV exposure status was not associated with early infant mortality (aOR = 1.06, 95% CI: 0.56–1.99).

## Mortality and HIV-free survival among HEI

Six infants were diagnosed with HIV infection by 6 weeks of age, including 4 diagnosed at birth, and 2 diagnosed at 6 weeks of age (Table 4). The estimated HIV transmission rate among those tested at birth was 0.9% (95% CI: 0.25–2.36) and by 6 weeks the overall HIV transmission was 1.0% (95%CI: 0.38–2.23). The estimated HIV-free survival including stillbirths was 92.8% (95% CI: 90.5–94.8), and 95.6% (95% CI: 93.7–97.1) when stillbirths were excluded.

**Table 2. Birth outcomes by mother's HIV status at delivery.**

| | Mother's HIV Status at delivery | | Total Mothers N = 1491 n (%) | P-value |
|---|---|---|---|---|
| | HIV-negative N = 868 n (%) | HIV-positive N = 623 n (%) | | |
| **Miscarriage*** | 7 (0.8) | 10 (1.6) | 17 (1.1) | 0.16 |
| **Mode of Delivery** | | | | |
| Vaginal | 748 (86.6) | 525 (84.8) | 1273(85.9) | 0.11 |
| Cesarean section | 116 (13.4) | 94 (15.2) | 210(14.1) | |
| **Place of delivery** | | | | |
| Health facility | 786/858 (91.6) | 560/620(90.3) | 1346(91.1) | |
| Home | 67 (7.8) | 53 (8.5) | 120 (8.1) | |
| Other | 5 (0.5) | 7 (1.1) | 12 (0.8) | |
| Missing data | 10 | 2 | 12 | |
| **Birth Outcome** | | | | |
| Liveborn† | 840 (96.8) | 603 (96.8) | 1443 (96.8) | |
| Antepartum death | 5 (0.6) | 12 (1.9) | 17 (1.1) | 0.01 |
| Intrapartum death | 23 (2.6) | 8 (1.3) | 31 (2.1) | |
| **Newborn Maturity** | | | | |
| Mature | 835 (96.0) | 574 (91.7) | 1409 (94.2) | 0.001 |
| Premature | 35 (4.0) | 52 (8.3) | 87 (4.8) | |
| Very Premature delivery (<32 wks) | 3 (0.4) | 13 (2.2) | 16 (1.1) | 0.001 |
| Premature delivery (<37 wks) | 31 (3.6) | 48 (7.8) | 79 (1.6) | <0.01 |
| Missing data | 5 | 1 | 6 | |
| **Newborn with congenital anomalies** | 9 (1.0) | 4 (0.6) | 13 (0.9 | 0.56 |
| **Birth Weight in kilograms** | | | | |
| Normal weight | 736 (90.8) | 514(88.3) | 1250(89.7) | 0.15 |
| Low Birth Weight (<2.5 kg) | 75 (9.2) | 68(11.7) | 143(10.3) | |
| Very Low Birth Weight (<1.5 kg) | 7 (0.9) | 8 (1.4) | 15(1.1) | 0.43 |

*There were no miscarriages or stillbirths recorded among the 8 women who seroconverted between enrollment and delivery

†This does not include babies who were born alive and died with two hours.

Five of the 6 HIV-infected infants had mothers that were initiated on ART post-conception (Table 5). Five of the mothers of infected infants had records of viral load at delivery and of these, 4 women had viral loads above 100,000 copies/ml. All mothers were on a TDF/3TC/EFZ regimen as per the national guidelines.

At 6 weeks, the mortality rate was higher among premature babies 24.6% (17/69) compared to mature infants 2.1% (29/1391). There was also a difference in 6-8-week mortality between low birth weight infants compared to normal birth weight infants (7.0% vs. 1.9%). Very low birth weight infants had higher risk of death within 6–8 weeks compared to infants born weighing at least 1.5kgs or more (30% vs 2.2%).

## Discussion

This is the first prospective cohort study in Lesotho comparing birth and 6-week outcomes between HEI and HUI in a large cohort of HIV-positive and negative women and their infants. We found that birth outcomes of HEI were similar to those of HUI except for the frequency of prematurity, which was found to be significantly higher among HEI. While prematurity among HEI has been reported in several other studies [17–19], there have been few studies

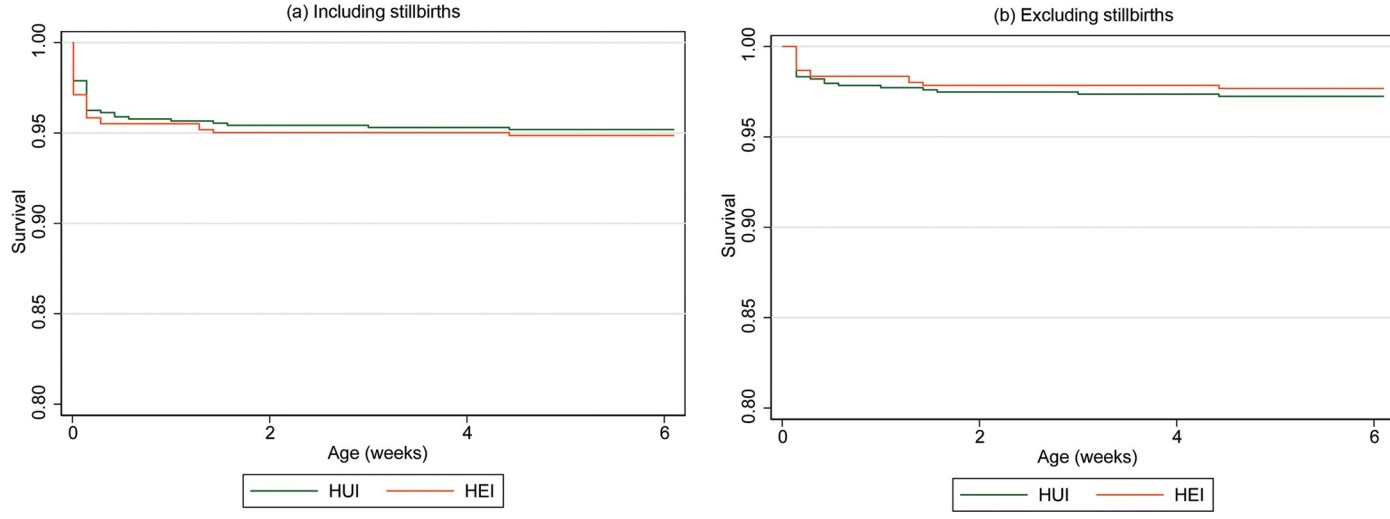

**Fig 2. Survival of HUI and HEI at six weeks of age including and excluding stillbirths.**

conducted within routine health systems with large numbers of both HIV-positive and HIV-negative women in the era of lifelong ART for all pregnant women with TDF-based ART regimens. In our study, infant prematurity and low birth weight were not significantly different among women who initiated ART before conception compared to those who initiated ART after conception. This is consistent with other studies that reported no relationship between preconception ART and preterm delivery [17,18]. However, in a systematic review and meta-analysis of adverse pregnancy outcomes and timing of initiation of ART, Uthman et. al reported significantly higher risk of prematurity among HEI whose mothers initiated ART before conception compared to those who initiated ART after conception (pooled RR 1.20, 95%CI 1.01–1.44) [8].

As recommended for first-line treatment in Lesotho, 88% of HIV-positive women were on TDF-based ART regimens. In a systematic review and meta-analysis looking at safety of TDF-based regimens in pregnancy for HIV-positive women and their infants, the rates of prematurity and stillbirths were significantly lower among women on TDF-based ART compared to other ART regimens [20, 21]. However, even with the use of TDF regimens, our study showed that HEI still had a higher risk of prematurity compared to HUI.

We found that HEI were more likely to die antepartum (1.9% versus 0.6%), consistent with medical complications, while HUI were more likely to die during the intrapartum period (2.6% versus 1.3%), consistent with obstetrical complications. However, it may be important to note that these women were followed up only from their first ANC and we may have missed some of the antepartum deaths which occurred before women were enrolled in the study. A number of studies have explored causes of antepartum death among HEI. In South Africa and Botswana, maternal vascular malperfusion was more frequent among HIV-positive women

**Table 3. Survival of HUI and HEI at six weeks of age.**

| | HIV-Unexposed | | HIV-Exposed | |
|---|---|---|---|---|
| | **Death (rate)** | **Survival (95% CI)** | **Death (rate)** | **Survival (95% CI)** |
| **Including stillbirths** | 46/877 (5.2%) | 94.8% [93.1–96.1] | 38/631 (6.0%) | 94.0% [91.8–95.7] |
| **Excluding stillbirths** | 27/858 (3.2%) | 96.8% [95.4–97.9] | 20/613 (3.3%) | 96.7% [95.0–98.0] |

**Table 4. HIV-free survival at six weeks of age.**

| | HIV-Exposed Infants | | | |
|---|---|---|---|---|
| | Death (rate) | HIV infection (rate) | Infected/Death (rate) | HIV-Free Survival (95% CI) |
| **Including stillbirths** | 38/627 (6.1%) | 6/581 (1.0%) | 44/613 (7.2%) | 92.8% [90.4–94.7] |
| **Excluding stillbirths** | 20/609 (3.3%) | 6/581 (1.0%) | 26/595 (4.4%) | 95.6% [93.7–97.1] |

and placenta insufficiency associated with hypertension accounted for most stillbirths [22,23]. In our study, postpartum infant mortality at 6 to 8 weeks was independently associated with gestational age, but not with HIV exposure status.

We found one of the lowest 6-week HIV transmission rates (1.0%) reported in Lesotho coupled with very high HIV-free survival among liveborn infants at 6 weeks of age. This is a significant improvement compared to the estimated 6-week transmission of 7% reported in Lesotho in 2016 and in other countries in the region [1, 24]. Two-thirds of the HIV-positive infants identified were infected in utero. Most women presented for their first ANC visit toward the end of the second trimester, contrary to the WHO recommendation for women to present during the first trimester. Early ANC visits are especially important for HIV-positive pregnant women because earlier ART initiation may further reduce MTCT in utero [12,25]. This was buttressed by the findings that almost all women who transmitted HIV to their infants had a high viral load despite being on ART. Therefore, we agree with Smith et. al that it is essential to maximize viral suppression for HIV-positive women on ART [25].

Our findings indicate that the current universal ART program within the setting of routine care is effective. Implementation of this program showed that high ANC utilization, and high uptake of ART during pregnancy, including a high proportion of facility-based deliveries, collectively led to improved pregnancy outcomes among HIV-positive women. Our study also demonstrates the importance of incorporating implementation research to document program effectiveness within routine, public health settings. Lesotho's experience shows that when PMTCT programs are well implemented, routine program setting can achieve high effectiveness, comparable to more controlled research settings.

A limitation of our study is that it measured birth outcomes and 6-week HIV-free survival in a facility-based population and may have missed women and children who did not seek care in health facilities. We may have also missed women who lost pregnancies early. In addition, our study may have potential systematic errors arising from estimation of gestational age with the potential of misclassifying baby's maturing at delivery. However, since ANC attendance in Lesotho is higher than in many African countries, we believe that the results are

**Table 5. Timing of ART initiation and maternal viral load at delivery for HIV-infected infants.**

| Infant # | Maternal age (years) | Gestational age at first ANC (weeks) | Timing of ART initiation | Maternal duration on ART before delivery (months) | Infant maturity and weight (kg) at birth | Maternal ART regimen at enrollment + | Maternal VL at delivery (copies/ml) |
|---|---|---|---|---|---|---|---|
| 1 | 23 | 32 | Pre-conception | 32.5 | Mature—2.4 | TLE | 36,881 |
| 2 | 22 | 24 | Post-conception | 3.5 | Mature—3.5 | TLE | 109,000 |
| 3 | 32 | 18 | Post-conception | 5.6 | Mature—3.5 | TLE | 428,054 |
| 4 | 18 | 18 | Post-conception | 4.6 | Mature—2.8 | TLE | 320,000 |
| 5 | 23 | 10 | Post-conception | 6.9 | Mature—2.8 | TLE | 100,062 |
| 6 | 25 | 29 | Post-conception | 2.6 | Mature—2.9 | TLE | - |

+All mothers were initiated on Tenofovir-Lamivudine-Efavirenz (TLE) after HIV diagnosis and remained on this regimen throughout the study.

reflective of the Lesotho context [26, 27]. The successful reduction of perinatal HIV transmission that we report may not be comparable in areas with lower ANC attendance and low rates of facility-based deliveries. Additionally, study sites were purposively selected so the findings may not be generalizable to the whole country of Lesotho, especially to low volume facilities, which were not included in the study. However, the purposive selection of sites from the three geo-ecological settings in Lesotho (highlands, foothills, lowlands) does account for the variances in health-seeking behaviors.

## Conclusion

Implementation of universal maternal ART was associated with low MTCT among infants at 6 weeks of age with no differences in congenital anomalies or early mortality between HEI and HUI. However, HEI continue to have increased rates of prematurity even in the era of lifelong combination ART.

## Supporting information

**S1 File. S3_File.excel birth outcomes data set.**
(XLSX)

## Acknowledgments

This study would not have been possible without the hard work and dedication of the entire study team and the women and children in Lesotho who participated in the study.

We also appreciate the support of the Lesotho Ministry of Health and the entire EGPAF Lesotho team.

This work was made possible by the United States Agency for International Development (USAID) and the generous support of the American people through USAID Cooperative Agreement Number 674-A-00-10-00031-00 and No. AID-674-A-16-00005. The content included here is the responsibility of the authors and does not necessarily represent the official views of these donors.

## Author Contributions

**Conceptualization:** Appolinaire Tiam, Seble G. Kassaye, Rhoderick Machekano, Michelle M. Gill, Majoalane Mokone, Mosilinyane Letsie, Mots'oane Tsietso, Irene Seipati, Anthony Isavwa, Laura Guay.

**Data curation:** Appolinaire Tiam, Seble G. Kassaye, Rhoderick Machekano, Vincent Tukei, Michelle M. Gill, Janety Barasa, Thorkild Tylleskär, Laura Guay.

**Formal analysis:** Appolinaire Tiam, Rhoderick Machekano.

**Funding acquisition:** Appolinaire Tiam.

**Methodology:** Appolinaire Tiam, Seble G. Kassaye, Rhoderick Machekano, Vincent Tukei, Michelle M. Gill, Majoalane Mokone, Mosilinyane Letsie, Mots'oane Tsietso, Irene Seipati, Anthony Isavwa, Thorkild Tylleskär, Laura Guay.

**Project administration:** Appolinaire Tiam, Vincent Tukei, Laura Guay.

**Resources:** Appolinaire Tiam, Majoalane Mokone.

**Software:** Rhoderick Machekano, Anthony Isavwa.

**Supervision:** Appolinaire Tiam, Vincent Tukei, Michelle M. Gill, Majoalane Mokone, Mot-s'oane Tsietso, Irene Seipati, Janety Barasa, Laura Guay.

**Writing – original draft:** Appolinaire Tiam.

**Writing – review & editing:** Seble G. Kassaye, Rhoderick Machekano, Vincent Tukei, Michelle M. Gill, Majoalane Mokone, Mosilinyane Letsie, Mots'oane Tsietso, Irene Seipati, Janety Barasa, Anthony Isavwa, Thorkild Tylleskär, Laura Guay.

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
