## [Decision Letter · Decision Letter 0]

12 Aug 2019

PONE-D-19-18206

Comparison of 6-week PMTCT outcomes for HIV-exposed and HIV-unexposed infants in the era of lifelong ART: results from an observational prospective cohort study

PLOS ONE

Dear Dr. Tiam,

Thank you for submitting your manuscript to PLOS ONE. After careful consideration, we feel that it has merit but does not fully meet PLOS ONE’s publication criteria as it currently stands. Therefore, we invite you to submit a revised version of the manuscript that addresses the points raised during the review process.

We would appreciate receiving your revised manuscript by Sep 26 2019 11:59PM. To enhance the reproducibility of your results, we recommend that if applicable you deposit your laboratory protocols in protocols.io, where a protocol can be assigned its own identifier (DOI) such that it can be cited independently in the future. For instructions see: http://journals.plos.org/plosone/s/submission-guidelines#loc-laboratory-protocols

We look forward to receiving your revised manuscript.

Kind regards,

Marcel Yotebieng

Academic Editor

PLOS ONE

2. We note that you have reported significance probabilities of 0 in places. Since p=0 is not strictly possible, please correct this to a more appropriate limit, eg 'p<0.0001'.

Additional Editor Comments (if provided):

We have heard back from reviewers. Both reviewers #1 and #3 provide very details suggestion on how to revised the manuscript and caution about the use of P-value, I particularly agree with Reviewer #1 suggestion to simply not use P-value.

Reviewers' comments:

Reviewer's Responses to Questions

**Comments to the Author**

1. Is the manuscript technically sound, and do the data support the conclusions?

Reviewer #1: Partly

Reviewer #2: Yes

Reviewer #3: Partly

2. Has the statistical analysis been performed appropriately and rigorously? 

Reviewer #1: No

Reviewer #2: Yes

Reviewer #3: No

3. Have the authors made all data underlying the findings in their manuscript fully available?

Reviewer #1: Yes

Reviewer #2: Yes

Reviewer #3: Yes

4. Is the manuscript presented in an intelligible fashion and written in standard English?

Reviewer #1: Yes

Reviewer #2: Yes

Reviewer #3: Yes

5. Review Comments to the Author

Reviewer #1: Question 1: I answered "partly" for the following reasons

a) I would recommend the authors define the variables collected further in the Methods section. As an example, was marital status self-report? Same for HIV status of their partner and disclosure. Specifically, what variables were from interviews and which were from medical records. This information will allow readers to better assess the results.

b) On lines 245-247, the authors mention HIV-exposed infants being more likely to die antepartum. While I don't disagree with the point, follow-up of women would need to begin at conception to make this determination. Rather only some antenatal period is captured (time since first ANC visit).

Question 2: I would have answered partly, but it wasn't an option. There are two points I would like to raise.

a) I would encourage the authors to reconsider their usage of p-values and statistical significance regarding interpretations. See the recent issues of The American Statistician. For the interpretation of p-values used by the authors, it requires that all error is random error. However, there is likely systematic error (confounding), since this is observational data and potential confounders are unaccounted for. The presence of the systematic error invalidates the interpretation of p-values used. It may be better to describe the general trends without an appeal to statistical significance.

Related, the systematic errors of the study should be discussed further in the limitations. While selection bias is discussed, there remains measurement error (newborn maturity is difficult to measure), missing data (women/infants lost to follow-up), and confounding (differences in maturity between HIV-exposed and HIV-unexposed may be due to some common cause).

b) Have the authors considered using Kaplan-Meier or similar estimators to report a survival curve instead? This would convey more information than survival at a single time point. It would also clarify the time-scale of interest (time since birth or time since first ANC visit)

Some additional minor comments:

- I am confused by the numbers on Line 138-141. The second and third/fourth sentence of the paragraph seem to repeat the same information. Furthermore, both n=652 and n=631 are said to be the number of HIV-exposed infants. It is not clear to me why these numbers are different.

- I would encourage the authors to write out HIV-exposed and HIV-unexposed infants rather than use abbreviations.

- Line 178, the p-value does not match Table 2.

- Why were both means and standard deviation, and median and interquartile range used? Based on the Results, it seems that only mean and standard deviations were used, but both are mentioned in the Methods.

Reviewer #2: Great manuscript - well written.

Minor detail: table 4 - the author uses (*) in the table and (+) in the footnote

Reviewer #3: Thank you for the opportunity to review this manuscript that presents a prospective cohort study of women living with (N=653) and without (N=941) HIV and their infants up to 6 weeks of age in Lesotho receiving care within routine health services. The study compared birth outcomes (stillbirth, preterm birth, low birth weight, congenital anomalies) and survival at 6 weeks of age in HIV exposed and HIV unexposed infants and also described HIV acquisition by 6 weeks of age in HIV exposed infants. The study found substantially higher rates of preterm and very preterm birth in HEI compared to HUI and a remarkably low rate of HIV-acquisition of 1% in HEI at 6 weeks of age.

Overall it is a valuable paper documenting the experience in Lesotho. I recommend however some re-consideration of the interpretation of the results and analyses presented. Please see detailed comments below.

Major comments:

• I would be cautious to make any conclusions about difference (present or absent) in congenital anomalies – the proportions were 0.6% in HUI and 1.0% in HEI, and although this isn’t a statistically significant difference in this sample, these are rare outcomes with low rates and the study is underpowered to detect this difference, the point estimate of which is approximately a 50% increased prevalence in HEI compared to HUI.

• Similarly for very low birth weight, by the point estimates there is an approximately 50% increased risk for VLBW in HEI compared to HUI, but the proportions are very low and thus this sample under-powered to compare these outcomes.

• Methods

o Please can you add to the methods section how gestational age was determined and what the definitions for preterm and very preterm birth were (later in the results the definition of very premature birth is given as < 32 weeks). Would also suggest using the standard terminology of ‘preterm birth’ rather than ‘premature birth’.

o Please also add to methods how miscarriages and stillbirths were defined

• The rate of miscarriage is extremely low in both groups (up to 20% of pregnancies end in miscarriage) – this may be due to how miscarriages were defined or more likely due to under-ascertainment, with likely only late miscarriages documented in this sample of women who largely presented for antenatal care well after the first trimester. For these reasons I would be hesitant to present results on miscarriages at all.

• The results presented illustrate very well that there is a pathway between HIV exposure, preterm birth and early mortality – HIV exposure is associated with a higher rate of preterm birth and preterm birth is associated with early mortality. I was therefore confused by the very final sentence of the results briefly presenting the only multivariable model described in the paper, that evaluated the association between preterm birth and early mortality adjusted for HIV exposure. From the earlier results presented preterm birth is on the causal pathway between HIV exposure (HIV exposure � preterm birth � early infant mortality) and thus adjusting for either HIV exposure or preterm birth in the reciprocal association with mortality is inappropriate. Lines 250-251 in the discussion that lead from this, discount that HIV exposure is still problematic as this is the risk factor for the increased preterm birth that is leading to the increased mortality.

• From the descriptive results presented there are additional potential maternal confounders to take into consideration when evaluating HIV exposure and early infant mortality – particularly the differences in maternal age and gestation at first ANC. Were additional multivariable analyses conducted to take these differences into account.

• Was morbidity evaluated in any way – e.g. hospitalization before 6 weeks of age? There may be improving survival but with the higher rate of preterm birth there may still be a substantial hospitalization or other longer-term morbidity experienced by HEI (and the health care system) that is missing from this analysis. If not evaluated, this should be added as a limitation.

Minor comments:

• Would you consider using alternative terminology in place of “mother-to-child transmission”? Women with HIV have repeatedly expressed finding this term stigmatizing and prefer alternatives such as vertical HIV transmission or perinatal and postnatal HIV transmission. I understand that most country program names are still called PMTCT Programs, and this may be difficult to avoid, however in all other contexts alternatively terminology would be possible.

• Generally, it is also now appropriate to use people-first terminology and refer to women living with/without HIV rather than HIV positive/negative women.

Introduction

• If PMTCT is to be used, please write out in full when used the first time in line 44.

• Line 52-53 seems incomplete at the end “Some studies found that more adverse birth outcomes such as....[where more common/occurred more often?]”

Results

• Lines 140-142 (pg 8) Please can you explain the denominators for both the HEI and HUI groups. Why are the denominators in parentheses not the same as the number stated in the narrative text i.e Of the 631 HEI we had information on 95.4% (623/653) – why is the 653 different to the 631? Same for the HUI why is the 941 different to the 879 HUI?

• Thank you for the contribution of distinguishing macerated from fresh stillbirths. This additional classification of stillbirths has seldom been given in the HIV birth outcomes literature and this distinction helps to start to understand potential mechanistic pathways.

• Table 4 – column heading “Gestational Age” is unclear – gestational age when (at delivery, at ART initiation, at first presentation for ANC?)

6. PLOS authors have the option to publish the peer review history of their article (what does this mean?). If published, this will include your full peer review and any attached files.

Reviewer #1: No

Reviewer #2: No

Reviewer #3: Yes: Amy L. Slogrove

---

## [Author Response · Author response to Decision Letter 0]

27 Sep 2019

PONE-D-19-18206

Comparison of 6-week PMTCT outcomes for HIV-exposed and HIV-unexposed infants in the era of lifelong ART: results from an observational prospective cohort study

General Comments from the Academic Editor

We note that you have reported significance probabilities of 0 in places. Since p=0 is not strictly possible, please correct this to a more appropriate limit, eg 'p<0.0001'.

Thank for the observation, we have made the necessary correction throughout the manuscript.

We have heard back from reviewers. Both reviewers #1 and #3 provide very details suggestion on how to revise the manuscript and caution about the use of P-value, I particularly agree with Reviewer #1 suggestion to simply not use P-value. 

 Thank you for the observation, we have taken note and have adjusted the text accordingly.

Review Comments to the Author

Reviewer #1: 

Comment 1: 

Question 1- I answered "partly" for the following reasons

a) I would recommend the authors define the variables collected further in the Methods section. As an example, was marital status self-report? Same for HIV status of their partner and disclosure. Specifically, what variables were from interviews and which were from medical records. This information will allow readers to better assess the results.

Thank you for the recommendation to clarify data collection methods for the different variables. We have amended the data collection section to indicate which variables were collected through structured interviews with pregnant women and variables extracted from medical charts and clinic registers.

b) On lines 245-247, the authors mention HIV-exposed infants being more likely to die antepartum. While I don't disagree with the point, follow-up of women would need to begin at conception to make this determination. Rather only some antenatal period is captured (time since first ANC visit).

Thank you for this important comment. We have added a sentence and the paragraph now reads as follows:

We found that HEI were more likely to die antepartum (1.9% versus 0.6%), consistent with medical complications, while HUI were more likely to die during the intrapartum period (2.6% versus 1.3%), consistent with obstetrical complications. However, it may be important to note that these women were followed up only from their first ANC visit and we may have missed some of the antepartum deaths which occurred before women were enrolled in the study.

Comment 2: 

Question 2- I would have answered partly, but it wasn't an option. There are two points I would like to raise.

a) I would encourage the authors to reconsider their usage of p-values and statistical significance regarding interpretations. See the recent issues of The American Statistician. For the interpretation of p-values used by the authors, it requires that all error is random error. However, there is likely systematic error (confounding), since this is observational data and potential confounders are unaccounted for. The presence of the systematic error invalidates the interpretation of p-values used. It may be better to describe the general trends without an appeal to statistical significance.

Related, the systematic errors of the study should be discussed further in the limitations. While selection bias is discussed, there remains measurement error (newborn maturity is difficult to measure), missing data (women/infants lost to follow-up), and confounding (differences in maturity between HIV-exposed and HIV-unexposed may be due to some common cause).

We take note of the reviewer’s concern on the p-value interpretations and the existence of systematic error due to potential measurement error and confounding. We have adjusted the text accordingly and acknowledge potential biases from the systematic errors.

b) Have the authors considered using Kaplan-Meier or similar estimators to report a survival curve instead? This would convey more information than survival at a single time point. It would also clarify the time-scale of interest (time since birth or time since first ANC visit)

We have now included Kaplan-Meier as figure 2 to report survival.

Comment 3:

I am confused by the numbers on Line 138-141. The second and third/fourth sentence of the paragraph seem to repeat the same information. Furthermore, both n=652 and n=631 are said to be the number of HIV-exposed infants. It is not clear to me why these numbers are different.

Thank you for the observation. We have corrected the numbers and the paragraph now reads as follows:

A total of 1594 pregnant women (941 HIV-negative and 653 HIV-positive) were enrolled in the study with their infants (Figure 1). Eight HIV negative women seroconverted before delivery. Delivery information was available for 95.4% of HIV positive women (623/653 and 92.2% of HIV negative women (868/941). 623 HIV positive women gave birth to 631 HIV exposed infants (HEI) and 868 HIV negative women gave birth to 879 HIV unexposed infants (HUI). Six-week follow up information was available for 577 and 831 HEI and HUI respectively.

Comment 4: 

I would encourage the authors to write out HIV-exposed and HIV-unexposed infants rather than use abbreviations.

We accept the recommendation.

Comment 5:

Line 178, the p-value does not match Table 2.

Thank you for identifying error. We have corrected the p-value to match the value in Table 2 which is the correct value.

Comment 6:

Why were both means and standard deviation, and median and interquartile range used? Based on the Results, it seems that only mean and standard deviations were used, but both are mentioned in the Methods.

We have edited the method to remove median and interquartile range. Indeed, the results reported on means and standard deviation.

Reviewer #2: 

Comment 7:

Great manuscript - well written.

Thank you very much for your kind words.

Comment 8:

Minor detail: table 4 - the author uses (*) in the table and (+) in the footnote

This has been corrected.

Reviewer #3: 

General Comment

Thank you for the opportunity to review this manuscript that presents a prospective cohort study of women living with (N=653) and without (N=941) HIV and their infants up to 6 weeks of age in Lesotho receiving care within routine health services. The study compared birth outcomes (stillbirth, preterm birth, low birth weight, congenital anomalies) and survival at 6 weeks of age in HIV exposed and HIV unexposed infants and also described HIV acquisition by 6 weeks of age in HIV exposed infants. The study found substantially higher rates of preterm and very preterm birth in HEI compared to HUI and a remarkably low rate of HIV-acquisition of 1% in HEI at 6 weeks of age.

Overall it is a valuable paper documenting the experience in Lesotho. I recommend however some re-consideration of the interpretation of the results and analyses presented. Please see detailed comments below.

Specific comments

Comment 9:

I would be cautious to make any conclusions about difference (present or absent) in congenital anomalies – the proportions were 0.6% in HUI and 1.0% in HEI, and although this isn’t a statistically significant difference in this sample, these are rare outcomes with low rates and the study is underpowered to detect this difference, the point estimate of which is approximately a 50% increased prevalence in HEI compared to HUI.

Thank you so much for your comment. We agree with you and have reviewed the text to read as follows.

The rate of congenital anomalies was 0.6% and 1.0% among HEI and HUI respectively.

Comment 10:

Similarly, for very low birth weight, by the point estimates there is an approximately 50% increased risk for VLBW in HEI compared to HUI, but the proportions are very low and thus this sample under-powered to compare these outcomes.

Thank you for the comment, we have edited the paragraph which now reads as follows:

The rates of very low weight (<1.5kgs) and low birth weight (<2.5 kgs) among women who initiated ART before conception was 1.7% and 11.6% respectively compared to 1.2% and 11.9% among women who initiated ART after conception.

Methods

Comment 11:

Please can you add to the methods section how gestational age was determined and what the definitions for preterm and very preterm birth were (later in the results the definition of very premature birth is given as < 32 weeks). Would also suggest using the standard terminology of ‘preterm birth’ rather than ‘premature birth’.

We have added the definitions and now use preterm birth as standard. Please see below.

Gestational age at birth was estimated by time between the date of last menstrual period given by the women at first ANC visit and the date of birth. Very preterm birth was defined as infant born at gestational age of 28-32 weeks while preterm birth was infant born after 32 weeks but before 37 weeks [15]

Comment 12:

 Please also add to methods how miscarriages and stillbirths were defined.

Thank you. The definitions have been added and reads as follows:

In addition, miscarriage was defined as loss of pregnancy before the gestational age of 28 weeks and stillbirth was considered when the pregnancy was lost after 28 weeks [16].

Comment 13:

The rate of miscarriage is extremely low in both groups (up to 20% of pregnancies end in miscarriage) – this may be due to how miscarriages were defined or more likely due to under-ascertainment, with likely only late miscarriages documented in this sample of women who largely presented for antenatal care well after the first trimester. For these reasons I would be hesitant to present results on miscarriages at all.

Thank you for the comment. We understand the concern of the reviewer, however it will be a missed opportunity not to present this important data point despite its limitation in terms of collecting data only for women who lost pregnancy after enrolment in the study. This is a limitation of the study because the study is taking place at health facility level. 

Comment 14:

The results presented illustrate very well that there is a pathway between HIV exposure, preterm birth and early mortality – HIV exposure is associated with a higher rate of preterm birth and preterm birth is associated with early mortality. I was therefore confused by the very final sentence of the results briefly presenting the only multivariable model described in the paper, that evaluated the association between preterm birth and early mortality adjusted for HIV exposure. From the earlier results presented preterm birth is on the causal pathway between HIV exposure (HIV exposure � preterm birth � early infant mortality) and thus adjusting for either HIV exposure or preterm birth in the reciprocal association with mortality is inappropriate. Lines 250-251 in the discussion that lead from this, discount that HIV exposure is still problematic as this is the risk factor for the increased preterm birth that is leading to the increased mortality.

Thank you for your comment. We agree with your observation and adjusting for HIV status is inappropriate as you rightly say. We have removed the HIV status adjusted result.

Comment 15:

From the descriptive results presented there are additional potential maternal confounders to take into consideration when evaluating HIV exposure and early infant mortality – particularly the differences in maternal age and gestation at first ANC. Were additional multivariable analyses conducted to take these differences into account.

While the distribution of both maternal age and gestational age at first ANC visit do differ by HIV exposure status, both maternal age and gestational age are not associated with early infant mortality therefore are unlikely to be confounders in the relationship between HIV exposure and early infant mortality. We went ahead and fitted a multivariate model of early infant mortality and HIV exposure adjusting for maternal age and gestational age. The unadjusted odds ratio for HIV exposure is 1.04 (95% CI: 0.58 – 1.87) while the adjusted odds ratio is 1.06 (95% CI: 0.56 – 1.99). We have included a statement in the results.

Comment 16:

Was morbidity evaluated in any way – e.g. hospitalization before 6 weeks of age? There may be improving survival but with the higher rate of preterm birth there may still be a substantial hospitalization or other longer-term morbidity experienced by HEI (and the health care system) that is missing from this analysis. If not evaluated, this should be added as a limitation.

We indeed did collected hospitalization data on a subset of the infants. However, there was no substantial difference in the rate of hospitalization between HIV exposed (N=532) and unexposed (N=589) infants in the first 6-8 weeks of life (1.32 % versus 1.02%).

Minor comments:

Comment 17:

Would you consider using alternative terminology in place of “mother-to-child transmission”? Women with HIV have repeatedly expressed finding this term stigmatizing and prefer alternatives such as vertical HIV transmission or perinatal and postnatal HIV transmission. I understand that most country program names are still called PMTCT Programs, and this may be difficult to avoid, however in all other contexts alternatively terminology would be possible.

Thank you for your comment. Although, recently the term HIV care for pregnant and breastfeeding women (PBFW) is being introduced, in the context of this paper, we felt mother to child transmission was still appropriate since the overall project was about evaluating effectiveness of PMTCT program.

Comment 18:

Generally, it is also now appropriate to use people-first terminology and refer to women living with/without HIV rather than HIV positive/negative women.

Thank you for this comment. This is very well noted and we agree with the suggestion. However, since this terminology is emerging, we will plead to use HIV negative and HIV positive terms here for consistency. 

Introduction

Comment 19:

If PMTCT is to be used, please write out in full when used the first time in line 44.

Thank you for the comment. This has been done.

Comment 20:

Line 52-53 seems incomplete at the end “Some studies found that more adverse birth outcomes such as....[where more common/occurred more often?]”

The sentence now reads:

Some studies found that adverse birth outcomes, such as increased preterm deliveries, stillbirths and low birth weight, occurred more frequently among HEI [3-8].

Results

Comment 21:

Lines 140-142 (pg 8) Please can you explain the denominators for both the HEI and HUI groups. Why are the denominators in parentheses not the same as the number stated in the narrative text i.e Of the 631 HEI we had information on 95.4% (623/653) – why is the 653 different to the 631? Same for the HUI why is the 941 different to the 879 HUI?

Thank you for the observation. We have corrected the numbers and the paragraph now reads as follows:

A total of 1594 pregnant women (941 HIV-negative and 653 HIV-positive) were enrolled in the study with their infants (Figure 1). Eight HIV negative women seroconverted before delivery. Delivery information was available for 95.4% of HIV positive women (623/653 and 92.2% of HIV negative women (868/941). 623 HIV positive women gave birth to 631 HIV exposed infants (HEI) and 868 HIV negative women gave birth to 879 HIV unexposed infants (HUI). Six-week follow up information was available for 577 and 831 HEI and HUI respectively. 

Comment 22:

Thank you for the contribution of distinguishing macerated from fresh stillbirths. This additional classification of stillbirths has seldom been given in the HIV birth outcomes literature and this distinction helps to start to understand potential mechanistic pathways.

Thank you for your kind comment.

Comment 23:

Table 4 – column heading “Gestational Age” is unclear – gestational age when (at delivery, at ART initiation, at first presentation for ANC?)

Thank you for the observation. This has been clarified in Table 4.

---

## [Decision Letter · Decision Letter 1]

22 Oct 2019

PONE-D-19-18206R1

Comparison of 6-week PMTCT outcomes for HIV-exposed and HIV-unexposed infants in the era of lifelong ART: results from an observational prospective cohort study

PLOS ONE

Dear Dr. Tiam,

Thank you for submitting your manuscript to PLOS ONE. After careful consideration, we feel that it has merit but does not fully meet PLOS ONE’s publication criteria as it currently stands. Therefore, we invite you to submit a revised version of the manuscript that addresses the points raised during the review process.

In addition to comments from the reviewer about overuse/interpretation of P-value, I also agree that further discussion of the systematic error will be to the benefit of this paper. 

We would appreciate receiving your revised manuscript by Dec 06 2019 11:59PM. To enhance the reproducibility of your results, we recommend that if applicable you deposit your laboratory protocols in protocols.io, where a protocol can be assigned its own identifier (DOI) such that it can be cited independently in the future. For instructions see: http://journals.plos.org/plosone/s/submission-guidelines#loc-laboratory-protocols

We look forward to receiving your revised manuscript.

Kind regards,

Marcel Yotebieng, M.D., MPH, Ph.D

Academic Editor

PLOS ONE

Reviewers' comments:

Reviewer's Responses to Questions

**Comments to the Author**

1. If the authors have adequately addressed your comments raised in a previous round of review and you feel that this manuscript is now acceptable for publication, you may indicate that here to bypass the “Comments to the Author” section, enter your conflict of interest statement in the “Confidential to Editor” section, and submit your "Accept" recommendation.

Reviewer #1: (No Response)

Reviewer #2: All comments have been addressed

Reviewer #3: All comments have been addressed

2. Is the manuscript technically sound, and do the data support the conclusions?

Reviewer #1: Yes

Reviewer #2: Yes

Reviewer #3: Yes

3. Has the statistical analysis been performed appropriately and rigorously? 

Reviewer #1: Yes

Reviewer #2: Yes

Reviewer #3: Yes

4. Have the authors made all data underlying the findings in their manuscript fully available?

Reviewer #1: Yes

Reviewer #2: Yes

Reviewer #3: Yes

5. Is the manuscript presented in an intelligible fashion and written in standard English?

Reviewer #1: Yes

Reviewer #2: Yes

Reviewer #3: Yes

6. Review Comments to the Author

Reviewer #1: I thank the authors for their revisions. I have a few minor comments that I believe will help improve the manuscript.

Line 48: MTCT is no longer abbreviated in the manuscript and therefore should be written out.

Lines 108-112: While the information is available in the tables, it may be useful to include the categories of responses in the text. Something along the lines of "HIV status of spouse (yes; no)".

Line 133: Only means are mentioned in the edited version but standard deviations are still calculated and presented in tables.

Lines 136-137: I see that the authors removed most of the p-values as previously suggested. Therefore, I am not sure why statistical tests (Chi-square and t-test/rank-sum) are discussed in the analysis section. Additionally, the authors may want to mention the use of the Kaplan-Meier estimator to generate the survival curves.

Lines 166, 168, 202: It appears that p-values are still used for interpreting some descriptive results.

Lines 239, 240, 242, 283: The word "significant" is used to describe the results. While maybe not what the authors intend, this may easily be misread by readers as statistically significant, so I would suggest replacing or removing the word "significant".

Lines 302-312: While the authors did add to their limitations, I believe that further discussion of systematic errors such as measurement error would be beneficial.

Reviewer #2: Minor comment: Line 26 in the abstract, the authors conclude that “Implementation of universal maternal ART lowers MTCT BY 6 weeks of age ...”. This conclusion is not supported by the findings. Did you mean “at 6 weeks of age? as is consistent with teh conclusion in the main paper (line 288)

Reviewer #3: Authors have addressed my initial comments adequately and I have no additional comments. Recommend manuscript for publication.

7. PLOS authors have the option to publish the peer review history of their article (what does this mean?). If published, this will include your full peer review and any attached files.

Reviewer #1: No

Reviewer #2: No

Reviewer #3: Yes: Amy L. Slogrove

---

## [Author Response · Author response to Decision Letter 1]

15 Nov 2019

Review Comments to the Author

Reviewer #1: 

I thank the authors for their revisions. I have a few minor comments that I believe will help improve the manuscript.

Line 48: MTCT is no longer abbreviated in the manuscript and therefore should be written out.

Thank you, MTCT has been written in full in the manuscript.

Lines 108-112: While the information is available in the tables, it may be useful to include the categories of responses in the text. Something along the lines of "HIV status of spouse (yes; no)".

The following sentence has been inserted: “Concerning HIV status of spouses, 4.2% of HIV negative women had HIV positive partners while 29.6% of HIV positive women had an HIV negative partner.”

Line 133: Only means are mentioned in the edited version but standard deviations are still calculated and presented in tables.

This has been corrected.

Lines 136-137: I see that the authors removed most of the p-values as previously suggested. Therefore, I am not sure why statistical tests (Chi-square and t-test/rank-sum) are discussed in the analysis section. Additionally, the authors may want to mention the use of the Kaplan-Meier estimator to generate the survival curves.

This has been corrected and the paragraph now reads as follows: “HIV-free survival was estimated as the proportion of children alive and HIV-negative among all exposed children. The precision around survival estimates was assessed by 95% confidence intervals. We used the Kaplan Meier curves to graphically display infant mortality, infection, and HIV free survival. We performed complete case analysis, and missing data were not imputed.”

Lines 166, 168, 202: It appears that p-values are still used for interpreting some descriptive results.

This has been corrected.

Lines 239, 240, 242, 283: The word "significant" is used to describe the results. While maybe not what the authors intend, this may easily be misread by readers as statistically significant, so I would suggest replacing or removing the word "significant".

The language has been corrected.

Lines 302-312: While the authors did add to their limitations, I believe that further discussion of systematic errors such as measurement error would be beneficial.

Thank you. We have added additional explanation.

Reviewer #2: Minor comment: Line 26 in the abstract, the authors conclude that “Implementation of universal maternal ART lowers MTCT BY 6 weeks of age ...”. This conclusion is not supported by the findings. Did you mean “at 6 weeks of age? as is consistent with the conclusion in the main paper (line 288)

Thank you. The sentence in the abstract has been corrected to align with the conclusion of the manuscript.

Reviewer #3: Authors have addressed my initial comments adequately and I have no additional comments. Recommend manuscript for publication.

Thank you very much.

---

## [Editor Report · Decision Letter 2]

26 Nov 2019

Comparison of 6-week PMTCT outcomes for HIV-exposed and HIV-unexposed infants in the era of lifelong ART: results from an observational prospective cohort study

PONE-D-19-18206R2

Dear Dr. Tiam,

We are pleased to inform you that your manuscript has been judged scientifically suitable for publication and will be formally accepted for publication once it complies with all outstanding technical requirements.

With kind regards,

Marcel Yotebieng, M.D., MPH, Ph.D

Academic Editor

PLOS ONE
---

## [Editor Report · Acceptance letter]

13 Dec 2019

PONE-D-19-18206R2 

Comparison of 6-week PMTCT outcomes for HIV-exposed and HIV-unexposed infants in the era of lifelong ART: results from an observational prospective cohort study 

Dear Dr. Tiam:

I am pleased to inform you that your manuscript has been deemed suitable for publication in PLOS ONE. Congratulations! Your manuscript is now with our production department. 

With kind regards,

on behalf of

Dr. Marcel Yotebieng 

Academic Editor

PLOS ONE